# Prenatal Diagnosis of Combined Maternal 4q Interstitial Deletion and Paternal 15q Microduplication

**DOI:** 10.3390/genes12101626

**Published:** 2021-10-16

**Authors:** Francesco Libotte, Marco Fabiani, Katia Margiotti, Antonella Viola, Alvaro Mesoraca, Claudio Giorlandino

**Affiliations:** 1ALTAMEDICA, Human Genetics Lab, Viale Liegi 45, 00198 Rome, Italy; francesco.libotte@artemisia.it (F.L.); marco.fabiani@artemisia.it (M.F.); antonella.viola@artemisia.it (A.V.); alvaro.mesoraca@artemisia.it (A.M.); claudio.giorlandino@artemisia.it (C.G.); 2Human Genetics Lab, Altamedica Main Centre, Viale Liegi 45, 00198 Rome, Italy; 3Fetal-Maternal Medical Centre, Department of Prenatal Diagnosis, Altamedica, Viale Liegi 45, 00198 Rome, Italy

**Keywords:** prenatal diagnosis, 4q deletion, array-CGH, variant of unknown significance, genetic counseling

## Abstract

The 4q deletion syndrome is a well-known rare genetic condition caused by partial, terminal, or interstitial deletion in the long arm (q) of chromosome 4. The phenotype of this syndrome shows a broad spectrum of clinical manifestations due to the great variability in the size and location of the deletion. In the literature, the mostly terminal deletions of chromosome 4q and the relative phenotypes are described, while the interstitial deletions of the long arm of chromosome 4 are rarely cited. Here, we report on a female fetus presenting no abnormal ultrasound evidence but with multiple chromosome aberrations. Comparative genomic hybridization (aCGH) revealed an interstitial 10.09 Mb deletion at the chromosome at the region of 4q28, arr[hg19] 4q28.1q28.3 (124068262_134158728)x1 combined with a 386.81 Kb microduplication at chromosome 15q11.1, arr[hg19] 15.11 (20249932_20636742)x3. At birth, and after 11 months, the baby was confirmed healthy and normal. The identification of this case allows for a deeper understanding of 4q syndrome and provides an explanation for the wide genetic/phenotypic spectrum of this pathology. This report can provide a reference for prenatal diagnosis and genetic counseling in patients who have similar cytogenetic abnormalities, and underlines the importance of reporting unusual variant chromosomes for diagnostic genetic purposes.

## 1. Introduction

The 4q deletion syndrome is a rare chromosomal disorder caused by a partial deletion of the long arm of chromosome 4. Its incidence has been estimated to be 1 in 100,000, with the majority of deletions occurring de novo and with approximately 14% of cases resulting from unbalanced segregation of the parental reciprocal translocations [1]. The term 4q deletion syndrome has been used to describe patients who have a deletion of the long arm of chromosome 4 as characterized by a standard karyotype analysis. The deletion of 4q31, 4q32, and 4q33-4qter leads to a distinctive syndrome with specific facial dysmorphism, cardiac and limb defects, and developmental delay, while more distal 4q deletion has been found in patients presenting with less severe symptoms [1]. The wide spectrum of phenotypes in 4q deletion syndrome patients has been described in correlation with the deletion size, the involved genes, and the genomic region. The deletion of the terminal region of the long arm of chromosome 4 was first recognized as a distinct syndrome by Townes [2], but only in 2009 did Wagner coin the term 4q syndrome for all cytogenetically visible deletions of the long arm of chromosome 4 [3]. The phenotype of this syndrome is well described in patients with a deletion of part or all the terminal third of the long arm in chromosome 4 [4]. In the literature, the majority of 4q deletions are terminal, whereas the interstitial deletions are less frequently reported and most of them have different breakpoints and a multitude of clinical phenotypes [5]. Chromosome analysis performed by array comparative genomic hybridization testing (aCGH) is often useful to refine the breakpoints identified by the conventional karyotype. In recent years, the number of submicroscopic chromosomal aberrations associated with 4q-phenotype has increased because of the widespread diagnostic use of aCGH [6]. In this study, we present a fetus with a 10.09 Mb deletion at the chromosome at the region of 4q28, arr[hg19] 4q28.1q28.3 (124068262_134158728)x1 combined with a chromosomal microduplication of 386.81 Kb at chromosome 15q11.1, arr[hg19] 15.11 (20249932_20636742)x3, and a reciprocal balanced translocation (9;22) (q34;q12) between chromosome 9 and 22. The aCGH analysis of the parental blood showed that the maternal 4 chromosome had the same 4q deletion whereas the paternal 15 chromosome had exactly the same 15q11.1 microduplication as the fetus. After genetic counseling, the parents decided to continue the pregnancy, and a female neonate was born on 26 November 2020 with no abnormal clinical manifestations having been observed to date. Although aCGH, or chromosomal microarray analysis (CMA), is a powerful diagnostic technology for detecting chromosomal copy number variations (CNVs), it often detects numerous variants of unknown significance (VUSs), thereby increasing the need for appropriate genetic counseling [7]. This study offers, for the first time, a clinical report on patients with chromosome 4q deletion involving bands q28.1q28.3 and 15q microduplication.

## 2. Case Presentation

A 37 year old pregnant woman primigravida, without ultrasound evidence and without a remarkable family history, came to our prenatal center for amniocentesis at 16 weeks of gestation due to maternal age and parental anxiety. Both parents were apparently normal. The result of the fetal karyotype by cytogenetic analysis indicated two structural chromosome anomalies—specifically, a reduction in the length of the long arm of one chromosome 4 with an anomalous banding pattern involving bands q28 (Figure 1 and Figure 2), and a further apparently balanced translocation between the long arm of chromosome 9 and chromosome 22 (Figure 1 and Figure 2).

To further investigate the specific position of the breakpoints and consequently involved genes in order to exclude a pathological phenotype, an aCGH was immediately performed on the fetal DNA. In parallel, to understand whether the abnormalities had de novo or parental origin, the cytogenetic analysis was extended to the prospective parents. The cytogenetic analysis on the parents revealed a normal paternal karyotype, whereas the maternal karyotype showed the same deletion in the 4q chromosome. In both parents, there was no evidence of translocation between chromosome 9 and 22, so it can be assumed that it is a de novo balanced translocation. The aCGH analysis performed on the fetal DNA revealed a 10.09 Mb interstitial deletion (chr4:124068262_134158728) within the region of 4q28.1q28.3, and a 380 kb microduplication (chr15: 20249932_20636742) within the region of 15q11.1 (Figure 1 and Figure 2). Then, to better understand the origin of the 15q11.1 microduplication, an aCGH analysis was also performed on the parents’ DNA. Interestingly, the molecular karyotype analyses of the parents showed that the paternal 15 chromosome had the same 15q11.1 microduplication as the fetus, while the maternal 4 chromosome had the 4q deletion already detected by the cytogenetic karyotype involving regions 4q28.1q28.3 as the fetus (Figure 3). The asymptomatic parents with the same karyotype led us to hypothesize that the maternal deletion can be considered a benign variant or a variant of unknown significance (VUS), while the paternal microduplication is a copy number variant (CNV). A VUS is a genetic variant for which, at the time of the interpretation, there was not sufficient evidence to determine whether the variant was related to disease or not.

A total of 11 genes were mapped in the 10.09 Mb deleted region of chromosome 4q28.1q28.3, among them 5 OMIM genes (online mendelian inheritance in Man) identified as the SPATA5, MFSD8, INTU, PLK4, and FAT4 genes. According to the OMIM database (www.omim.org, accessed on 6 June 2021), the spermatogenesis-associated protein 5 (SPATA5) is involved in spermatogenesis [8], but many studies have suggested a role in neuronal development [9]. Despite the multiple chromosomal abnormalities after the genetic counseling explaining all the potential risks, the couple decided to continue with the pregnancy. A follow-up of the pregnancy and the birth was performed, and there was no abnormality during the pregnancy nor during the birth, as the infant showed normal vital parameters. Birth weight, head circumference, and length were all within normal intervals, and no physical features were observed. After 7 months, the baby was found to show normal weight increase and normal length for her age; moreover, after the ultrasound imaging analysis and psychiatric evaluation, she has found to exhibit both normal bone and mental development.

## 3. Materials and Methods

Amniotic fluid was collected at 16 weeks of gestation. Cytogenetic analysis was performed on cultured amniocytes by G-banding according to standard procedures analyzing 16 metaphases. Chromosome analysis of parental blood samples was performed using GTG-banding techniques on PHA-stimulated blood lymphocytes. Array-CGH was performed on DNA from cultured amniocytes and DNA from parental blood to characterize the extent of deletion and microduplication, using 180 K platform (Agilent Technologies, Santa Clara, CA, USA). Briefly, 500 ng of the proband first, and of the parents later, with the relative sex-matched reference DNAs, was processed according to the manufacturer’s protocol. Fluorescence was scanned in a dual laser scanner (InnoScan 710, Agilent Technologies, Santa Clara, CA, USA), and images were extracted and analyzed via Agilent Feature Extraction Software. The position of oligomers refers to the human genome February 2009 (version GRCh37, hg19) assembly.

## 4. Discussion/Conclusions

Interstitial deletions of the long arm of chromosome 4 are rare, and most interstitial deletions involve the q11-q31 region. In our literature search, we found no reports describing any case with interstitial deletion of the long arm of chromosome 4, the microduplication of chromosome 15, and a de novo translocation (9;22). The 4q deletion syndrome is a distinct congenital malformation affecting multiple systems and organs, including facial and digital dysmorphology, autistic spectrum disorder, and abnormalities of the cardiovascular, musculoskeletal, and gastrointestinal system [10]. A broad spectrum of clinical manifestations has been observed partly due to the variability in the extent of the deletions and the possible additional contribution of other genetic rearrangements [11]. Here, we report a prenatal diagnosis of a particular case of 4q-interstitial deletion. The pregnant woman has normal ultrasound findings without a remarkable family history of genetic malformation, and who underwent amniocentesis at 16 weeks of gestation due to maternal age and parental anxiety. Both parents were apparently normal. After invasive testing, the conventional cytogenetics investigation led to the diagnosis of the interstitial 4q deletion at the region q28 and an apparently balanced translocation between the long arm of chromosome 9 and the long arm of chromosome 22 (Figure 1). In addition, the molecular analysis of cultured amniocytes with an aCGH further defined the chromosome 4 abnormality precise deletion breakpoints and deletion size, which are 4q28.1–28.3 and approximately 10 Mb (Figure 2). Moreover, the aCGH detected a 380 Kb balanced microduplication at the region of 15q11.1. Afterwards, a cytogenetic molecular analysis of both parents revealed that the deletion was inherited by the maternal parent and that translocation was a de novo chromosomal rearrangement. The aCGH of the parental blood confirmed maternal deletion with the same breakpoints, a balanced translocation, and highlighted the paternal microduplication at the region of 15q11 (as shown in Figure 2 and Figure 3). The final karyotype of the fetus was 46, XX, t(9;22)(q34;q12), del(4)(q28.1q28.3) mat. Unexpectedly, no reports about this loss were found in the Database of Genomic Variants (DGV) and no pathogenicity was reported in the Decipher database. A total of 11 genes, including SPATA5, MFSD8, INTU, PLK4, and FAT4 genes have been mapped to the 10.09-Mb deleted region of chromosome 4q28.1q28.3 (124068262–134158728). According to the online mendelian inheritance in Man (OMIM, www.omim.org, accessed on 6 June 2021), the spermatogenesis-associated protein 5 (SPATA5) is involved in spermatogenesis [8], but many studies have suggested a role in neuronal development [9]. All patients with SPATA5 variants reported in the literature so far have presented with developmental delay starting in early infancy, while 77% presented sensorineural hearing loss, 73% suffered from gastrointestinal problems, and 67% revealed an abnormal brain, including hypoplasia of the corpus callosum [12]. Fat atypical cadherin 4 (FAT4) is a key regulator of mammalian neurogenesis [13]. The inturned planar cell polarity protein (INTU) is known as a ciliogenesis and planar polarity effector (CPLANE) protein. Although the roles for INTU have been reported during embryonic development and in the context of developmental disorders, its function and regulation in adult tissues remain poorly understood [14]. Major facilitator superfamily domain-containing protein 8 (MFSD8) is one of the genes involved in neuronal ceroid lipofuscinoses (NCLs), commonly known as Batten disease, belonging to a family of neurological disorders that cause blindness, seizures, loss of motor function and cognitive ability, and premature death [15]. Polo-like kinase 4 (PLK4) was identified as a master regulator of centriole duplication. The loss of PLK4 or the inhibition of its kinase activity results in a failure to assemble procentrioles [16]. Based on what has been discussed above, the 10.09 Mb deletion of chromosome 4q28.1–28.3 is a VUS. In prenatal diagnosis, VUS pose a challenge for genetic counseling. The information obtained from prenatal diagnosis could facilitate prospective parents’ reproductive decisions when confronted with the choice between terminating and continuing the pregnancy. To clarify the clinical significance of the diagnosis, parental verification tests with aCGH of the parental blood showed a normal paternal 4 chromosome, whereas the maternal 4 chromosome had the same deletion at the 4q28.1–28.3 position, exactly superimposable with that of the fetus.

Unfortunately, we are unable to study the maternal or paternal parents in order to understand the origin of the abnormalities because both sets of parents had died of causes not attributable to genetic diseases. In the discussed case, three different chromosomal abnormalities were present, although two of them had been inherited from apparently healthy parents (4q interstitial deletion from the mother and 15q microduplication from the father) and one of them is a balanced translocation that is rarely pathogenic. Indeed, the deletion of the 4q chromosome could lead to 4q syndrome, which is associated with facial dysmorphism and/or other major malformations such as skeletal abnormalities and renal hypoplasia.

The present literature refers to several cases reporting 4q deletions, mostly terminal 4q deletion [1,17,18,19,20]. There are also a few cases of interstitial deletions that have been reported. A recent paper sharing a similar deletion as the present case showed a novel 7.22 Mb deletion at chromosome 4q32.2q32.3 (162858958–170081268), indicating that as a VUS [21]. Other authors have reported a 6 month old boy with a 4q31-q32 deletion who presented with congenital heart disease and clenched hands, and other authors [5] have reported a 33.5 Mb deletion of 4q32.1-q35.2 in a prenatal diagnosis presenting with high-risk combined screening test results and altered ultrasound markers. Aladhami et al. [22] reported a 12 year old boy and his mother both bearing the whole region of 4q32 and 4q3 and showing behavior problems, learning difficulties, and mild dysmorphic features [22]. We showed that the fetus inherited this 4q-deletion from an asymptomatic and healthy pregnant woman with no ultrasound anomalies. After genetic counseling, the couple decided to continue with the pregnancy, and the baby was born healthy without any abnormal traits attributable to genetic syndrome. Following the above considerations, we can assume that the 10.09 Mb deletion at chromosome 4q28.1q28.3 (124068262–134158728) could be considered a variant of uncertain significance (VUS) and not associated with any form of genetic disease even if combined with a microduplication at chromosome 15q11.1 or a balanced translocation (9;22) (q34;q12), as shown in the discussed case. The American College of Medical Genetics and Genomics (ACMG) guidelines agree that VUS should not be used in clinical decision making but informing about VUS is clinically relevant exclusively when, in the chromosomal affected region, the mapped genes are involved in pathogenic clinical traits. In conclusion, we can speculate that this variant could likely be classified as a benign variant even though the exclusion of the pathogenicity cannot be completely defined until a longer follow-up of the baby’s healthy status is carried out in order to exclude adult onset pathologies.

## Figures and Tables

**Figure 1 genes-12-01626-f001:**
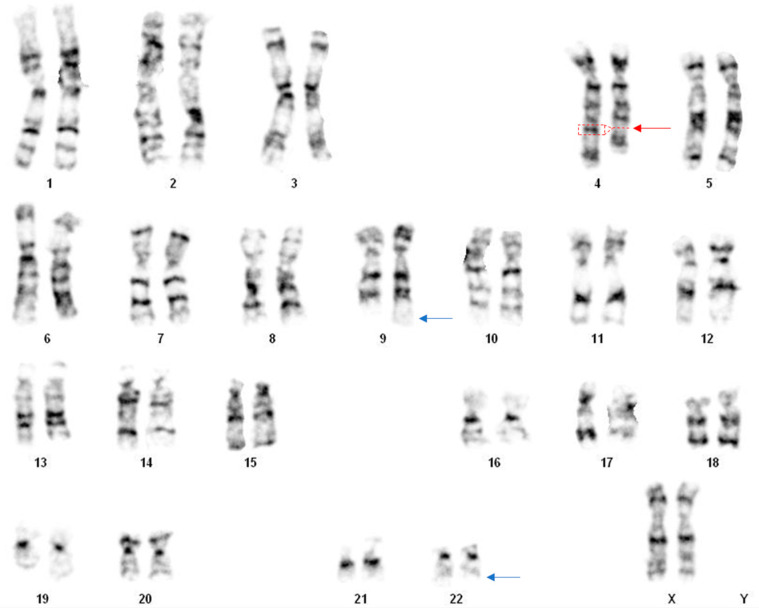
G-banded karyotype of the fetus indicated a 4q deletion (red arrow) and translocation (9;22) (blue arrows).

**Figure 2 genes-12-01626-f002:**
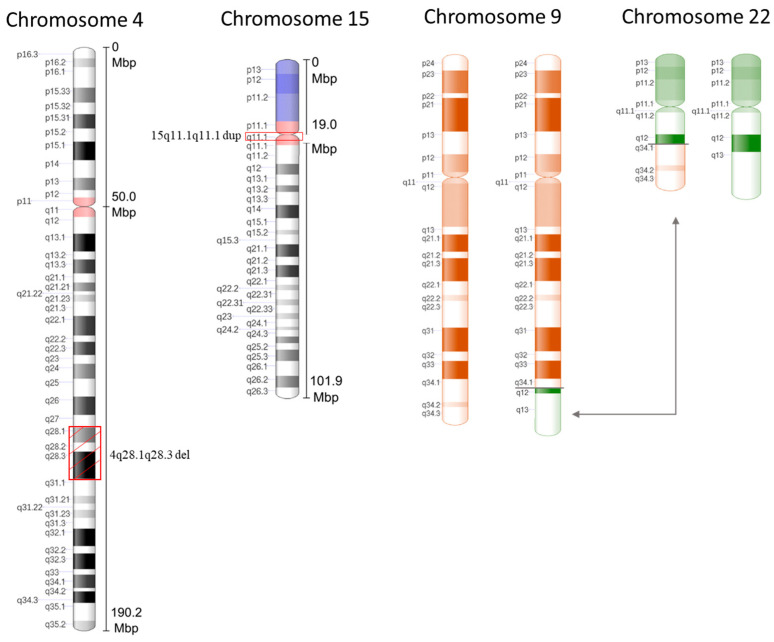
Ideograms of chromosome 4 with 4q28.1q28.3 deletion, chromosome 15 with 15q11.1q11.1 microduplication, and reciprocal translocation (9;22) (q34;q12) between chromosome 9 and 22.

**Figure 3 genes-12-01626-f003:**
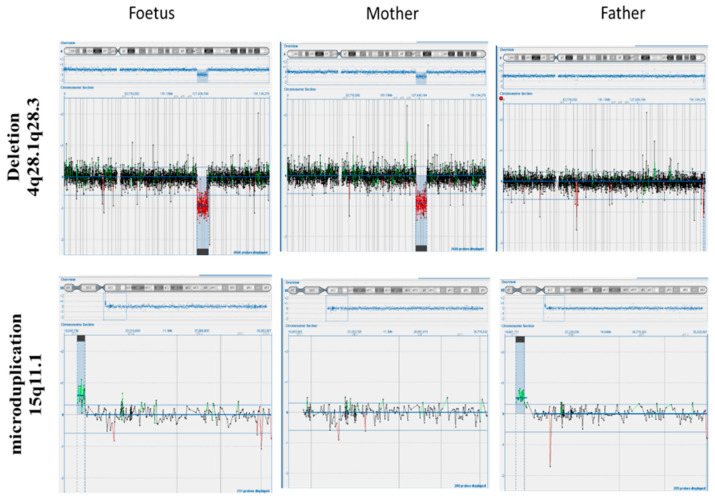
The array-comparative genomic hybridization (aCGH) result of the fetus, mother, and father. The size of deletion in chromosome 4 is estimated to be 10.09 Mb (from 124068262 kb to 134158728 kb). There are 11 OMIM deleted genes within the region with possible effects on the phenotypes: SPATA5; SPRY1; FAT4; INTU; SLC25A31; PLK4; MFSD8; PGRMC2; PHF17; SCLT1; PCDH10. The size of microduplication is estimated to be 380 Kb (from 20249932_20636742). There are no OMIM genes duplicated within the region. Size and position of deletion and microduplication are perfectly superimposable between parents and fetus.

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
