# Peer review of "Prenatal Diagnosis of Combined Maternal 4q Interstitial Deletion and Paternal 15q Microduplication"

_genes, 2021, doi:10.3390/genes12101626_

Round 1

Reviewer 1 Report

In this report, the authors present a fetus with novel variant of unknown significance (VUS) of 10.09 Mb deletion at chromosome 4q28.1q28.3, inherited from a phenotypically normal mother, a 386.81 Kb microduplication at chromosome 15q11.1, inherited from a phenotypically normal father, and a de novo translocation (9;22) (q34; q12).

My first observation is that the manuscript is poorly written and composed, it’s hard to follow at times and the logical flow is also often missing. The manuscript requires a complete revision. Complex sentences need rephrasing.

 Language/spell/grammar checking is required throughout the manuscript.

Major revisions

  • I suggest to rewrite the abstract according to the authors guideline showed in the journal website:

“A single paragraph of about 200 words maximum. For research articles, abstracts should give a pertinent overview of the work. We strongly encourage authors to use the following style of structured abstracts, but without headings.”

  • I suggest to improve the Introduction section adding some sentence to better clarify the aim of our manuscript.

  • Though the manuscript the authors discusses some interesting point of the research area but lacks in comprehensiveness and important insights from the authors.

  • In cases where prior experimental data are cited, require additional discussion from the authors as to why those observations are relevant (in the discussion section).

  • The author has provided interesting topic to start with, but it requires some more of their own critical input in the

  • I suggest to analyse also the parents of the pregnant woman investigated in order to better define the origin of the variation identified.

Minor revisions

  • I suggest to remove the sentence “A VUS is a rare or novel genetic chromosomal aberration for which the association with disease is unclear” from the introduction.
  • I suggest to write figure 1, figure 2 as suggested by authors guidelines
  •  

Reviewer 2 Report

The manuscript presents a case report about a female fetus, with no ultrasounds structural abnormalities, but with multiple chromosomes aberrations with unknown signification: a 10.09 Mb maternal interstitial deletion at chromosome 4q28.1q28.3, a paternal microduplication at chromosome 15q11.1q11.1, and a de novo balanced reciprocal translocation involving chromosomes 9 and 22. Standard karyotyping, following amniocentesis, revealed a deletion at band 4q28, a de novo translocation (9;22) (q34;q12), and a followed array Comparative Genomic Hybridization (aCGH) confirmed a 10.09 Mb deletion at chromosome region 4q28, arr [hg19] 4q28.1q28.3 (124068262_134158728)x1 showing a further chromosomal abnormality thus 386.81Kb microduplication at chromosome 15q11.1, arr [hg19] 15.11 (20249932_20636742)x3.

The case is very well presented, very well documented.

What is the usefulness of this diagnosis? I suggest to add information about ethical considerations. Many doctors agree that variants of unknown clinical significance should not be disclosed, and I suggest authors argue this.

Round 2

Reviewer 1 Report

The authors  answered all questions , the paper seems to fit with the proposal of the journal